# `AlphaVerus`: Bootstrapping Formally Verified Code Generation through Self-Improving Translation and Treefinement

**Pranjal Aggarwal** [1] **Bryan Parno** [1] **Sean Welleck** [1]

## Abstract

Automated code generation with large language models has gained significant traction, but there remains no guarantee of the correctness of generated code. We aim to use formal verification to provide mathematical guarantees that the generated code is correct. However, generating formally verified code with LLMs is hindered by the scarcity of training data and the complexity of formal proofs. To tackle this challenge, we introduce `AlphaVerus`, a self-improving framework that bootstraps formally verified code generation by iteratively translating programs from a higher-resource language and leveraging feedback from a verifier. `AlphaVerus` operates in three phases: exploration of candidate translations, Treefinement—a novel tree search algorithm for program refinement using verifier feedback, and filtering misaligned specifications and programs to prevent reward hacking. Through this iterative process, `AlphaVerus` enables the `LLaMA-3.1-70B` model to generate verified code without human intervention or model fine-tuning. `AlphaVerus` shows an ability to generate formally verified solutions for HumanEval and MBPP, laying the groundwork for truly trustworthy code-generation agents.[1]

## 1. Introduction

There has been an enormous effort to train code-generating large language models (LLMs) (Chen et al., 2021; Austin et al., 2021; Li et al., 2023; Rozière et al., 2024; Team, 2024), leading to LLM-powered agents that can perform tasks rang-

ing from fixing bugs in software repositories to solving Olympiad-level algorithmic problems (Jimenez et al., 2024; Li et al., 2022). Despite these successes, multiple studies have identified disturbing mistakes in LLM-produced code, including subtle bugs and serious security vulnerabilities (Hendler, 2023; Pearce et al., 2021; Jesse et al., 2023; Zhong & Wang, 2024; Perry et al., 2023; Elgedawy et al., 2024). Ultimately, these mistakes stem from a fundamental property of LLMs: language models can generate any string of code, without regard to correctness. As a result, automatically checking the correctness of LLM-generated code is one of the grand challenges facing the research community.

The generated code must be correct for all possible inputs it may receive. However, today's code generation methods select or filter generations with imperfect proxies of correctness, such as runtime testing or human inspection. Achieving perfect test coverage is typically infeasible (Li et al., 2022; Liu et al., 2023), and incomplete coverage leads to an unreliable signal that can be exploited by a model (Pan et al., 2022; Liu et al., 2023; Denison et al., 2024). Relying on human review is equally problematic since it scales poorly and humans can struggle to tell whether LLM-generated code is correct (Perry et al., 2023). In turn, the difficulty of trusting generated code reduces the potential productivity gains from using LLMs and can lead to unexpected vulnerabilities or unreliable signals for improving models.

In contrast, generating code in a *verification-aware programming language* such as Dafny (Leino, 2010), F* (Swamy et al., 2016), or Verus (Lattuada et al., 2023; 2024) offers a promising approach to addressing these challenges by providing mathematical guarantees that a program obeys a specification for all possible inputs. In this paradigm, code is paired with a specification and proof written in a specialized language, and a mechanical verifier checks whether the code meets the specification. This could dramatically improve the trustworthiness of the generated code: if the verifier passes, the LLM's generated program is mathematically guaranteed to meet the specification. However, writing formal specifications and proofs introduces additional layers of complexity. Furthermore, although LLMs have demonstrated success in automated theorem proving in mathematical domains (Lu et al., 2023; Li et al., 2024), their capability to generate

---

[1]Carnegie Mellon University. Correspondence to: Pranjal Aggarwal <pranjala@cmu.edu>, Bryan Parno <parno@cmu.edu>, Sean Welleck <wellecks@cmu.edu>.

*Proceedings of the $42^{nd}$ International Conference on Machine Learning*, Vancouver, Canada. PMLR 267, 2025. Copyright 2025 by the author(s).

[1]Code is available at https://alphaverus.github.io

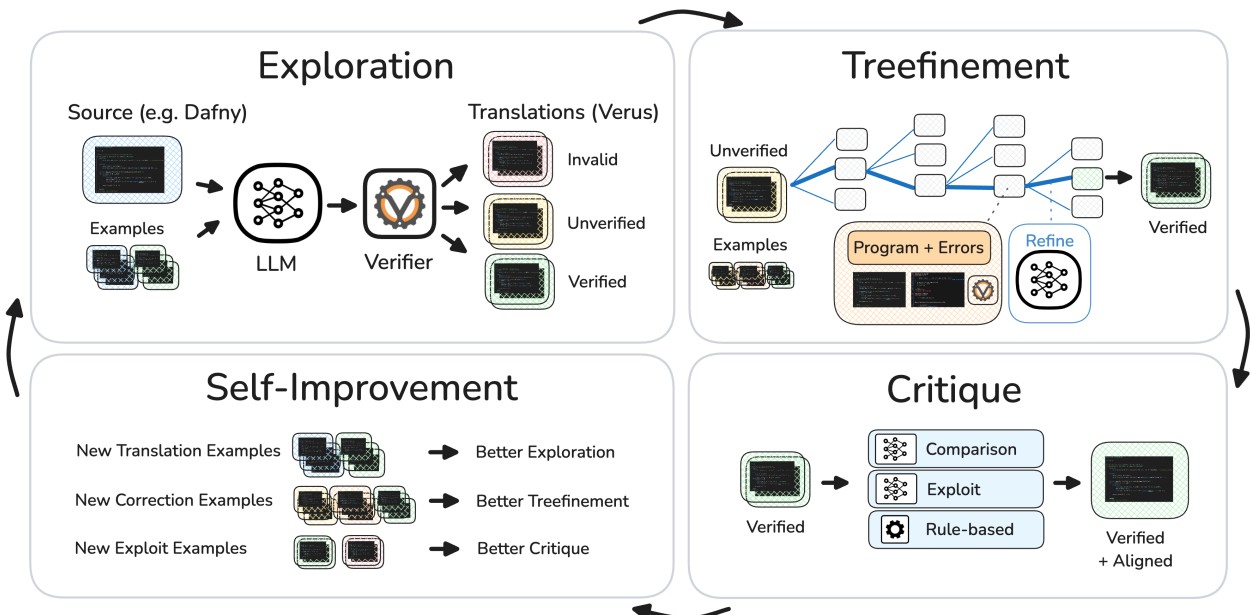

Figure 1: Overview of `AlphaVerus`, a self-improving framework for generating formally verified code. Each iteration consists of three key steps: (1) *Exploration* translates programs from a source language to Verus by sampling multiple trajectories and selecting partially correct ones using verifier feedback, (2) *Treefinement* iteratively fixes errors guided by verifier feedback and tree search, and (3) *Critique* validates and filters out underspecified or incorrect translations. The framework bootstraps new exemplars after each iteration to continuously improve performance without human intervention.

verified code for even basic algorithms is limited (Sun et al., 2024a; Lohn & Welleck, 2024).

A significant barrier to automatically generating real-world, formally verified code is the scarcity of training data. In particular, verification-aware research languages have a rich history (e.g., Dafny (Leino, 2010) or F* (Swamy et al., 2016)), yet verifying real-world code in mainstream languages remains nascent. For example, Verus (Lattuada et al., 2023)—a verification language for the very popular language Rust—has fewer than 10 public repositories, despite Rust itself having millions of code examples. Hence, enabling formally verified code generation in a mainstream language such as Rust faces a *bootstrapping problem*: without training data, how do we create an initial model that can generate even relatively simple verified programs?

We propose `AlphaVerus`, a framework for bootstrapping a formally verified code generation model by iteratively translating programs from a resource-rich domain and self-improving using feedback from the verifier. As illustrated in Figure 1, each iteration of `AlphaVerus` has three phases. First, the *exploration* phase generates candidate programs by translating from a source language (such as Dafny) to the target language (here, Verus) by generating multiple candidates and saving partially and completely verified attempts. Second, *Treefinement* refines the imperfect candidates through a novel tree search over the space of output programs using

feedback from the verifier, saving the final verified program, along with its ancestors to serve as error correction examples. We show that Treefinement leads to substantial gains over vanilla refinement strategies that resemble those used in concurrent work (Yang et al., 2024; Chen et al., 2024). Third, *critique models* detect misaligned translations and specifications–the one part of the pipeline that lacks formal guarantees. Crucially, this alleviates *reward hacking*, in which models learn to game the system by generating trivial or incomplete specifications, or even by identifying verifier limitations that cause trivial programs to pass the verifier. While previous work has investigated methods that rely on test cases (Sun et al., 2024a), our critique models address the challenging problems of automated specification generation and validation without relying on any unit test cases.

Each iteration of `AlphaVerus` collects new exemplars that improve the models in each phase, creating a cycle of improvement. Thus, unlike recent work that relies on human experts to write correction prompts (Yang et al., 2024), our method requires no human intervention and automatically learns to generate better code. Moreover, the system operates using a single language model (e.g., Llama 70b), without the need for the expensive GPT-4 initialization used in concurrent work (Chen et al., 2024). Finally, the collected exemplars can be used to improve the verified code generation performance of any model without any finetuning.

To demonstrate `AlphaVerus`, we consider Dafny (Leino, 2010) programs as the source domain, since the Dafny language has been around for over a decade and has accumulated a reasonable amount of code. We run `AlphaVerus` to automatically collect the DAFNY2VERUS-COLLECTION, a dataset of trajectories containing translated programs, error corrections, and critique examples based on the source dataset DafnyBench (Loughridge et al., 2025)–a dataset of 562 programs of varying difficulty. Finally, we evaluate the `AlphaVerus` pipeline by using the resulting data as few-shot exemplars for the downstream task of *formally verified code generation*: generating complete, formally verified implementations—including both algorithmic code and proof annotations—given human-written specifications Formally verified code generation is a significant step over concurrent work that focuses solely on the simplified, artificial setting of generating proof annotations for correct pre-written code (Yang et al., 2024; Chen et al., 2024). We show `AlphaVerus` enables Llama-70B to successfully generate verified solutions to 33% of HumanEval-Verified (The HumanEval-Verus Contributors, 2024), outperforming GPT-4o-based methods. Furthermore, through ablations, we establish the necessity of each component in `AlphaVerus`.

In summary, our contributions are five-fold: (1) We propose `AlphaVerus`, a novel self-improving framework for generating formally verified code; (2) We present a novel combination of tree search and refinement that improves over time; (3) We propose a novel critique phase, which has to our knowledge the first neural method that can improve the quality of specifications without test cases; (4) We introduce a new dataset containing formally verified Verus programs, along with error pairs; and (5) We demonstrate the effectiveness of our approach, evaluating its formally verified code generation abilities and ablating its components. In particular, `AlphaVerus` is the first method to achieve non-zero formally verified code generation performance on a verified version of HumanEval (Chen et al., 2021), thus establishing a starting point for code generation models that generate increasingly complex—yet trustworthy—code.

## 2. Formally Verified Code Generation

Our goal is to develop a model that generates formally verified code in a real-world programming language, which we refer to as *formally verified code generation*. Next, we provide background and then introduce `AlphaVerus`.

**Formal verification of code.** Formal verification ensures that a program adheres to a formally defined specification of its intended behavior. As illustrated in Figure 2, formally verified code typically consists of three components: (1) formal specifications $y_S$ defining the expected input-output behavior; (2) a code implementation $y_I$ intended to satisfy the specifications; and (3) a proof $y_P$ demonstrating that the

implementation conforms to the specifications. A verifier $v(y_S, y_I, y_P) \rightarrow \{0, 1\}$ uses the proof to statically check that the implementation meets the specification for all possible inputs, returning 1 if the program is correct with respect to $y_s$ and 0 if verification fails. Upon failure, the verifier additionally returns a set of messages $\{m_1, \ldots, m_M\}$ containing the number of verified statements, the number of errors, and localized error messages (e.g., see Figure 2).

**Misaligned specs and implementations.** The specifications themselves are not verified, as they represent the developer's intended behavior. Therefore, it is critical that the specifications accurately reflect the desired input-output behavior for all possible inputs. We use the term *misaligned* to refer to situations in which the specification does not reflect the desired input-output behavior. This includes misalignments between the specification and the developer's intent or the implementation, which can occur due to language features that cause programs to pass the verifier trivially (e.g., using "`assume(false)`").

**Formally verified code generation.** Our goal is to develop a model that generates formally verified code given a specification. Specifically, $(y_I, y_P) \sim G(y_S; c, \theta)$, where $G(\cdot)$ is a generation algorithm such as sampling from a language model with parameters $\theta$, and the model generates both an implementation $y_I$ and proofs $y_P$ given a specification $y_S$ and any additional context $c$. The goal is for the resulting code to verify, i.e., $v(y_S, y_I, y_P) = 1$.

**Bootstrapping formally verified code generation.** A practical goal is to perform formally verified code generation in a mainstream language, such as Rust code verified with the Verus verifier (Lattuada et al., 2023). However, doing so raises a technical challenge: it is infeasible to train a model on $(y_S, y_I, y_P)$ examples since few examples exist. We refer to this as a *bootstrapping problem*, since we need to create an initial generation model (that we may subsequently improve) without any training data. Next, we describe `AlphaVerus`, a framework for bootstrapping a verified code generation model by translating from a more resource-rich language.

## 3. `AlphaVerus`

To generate verified code in the absence of training data in our target language (Verus), we propose to iteratively translate programs from a higher-resource domain into Verus. Each iteration collects data by exploring candidate translations, refining them with a novel tree search, and then filtering out misaligned programs. Finally, we use the data to enable a verified code generation model (via few-shot learning), and evaluate the model plus the tree search on the downstream task of verified code generation: generating verified code and proofs given a held-out test specification.

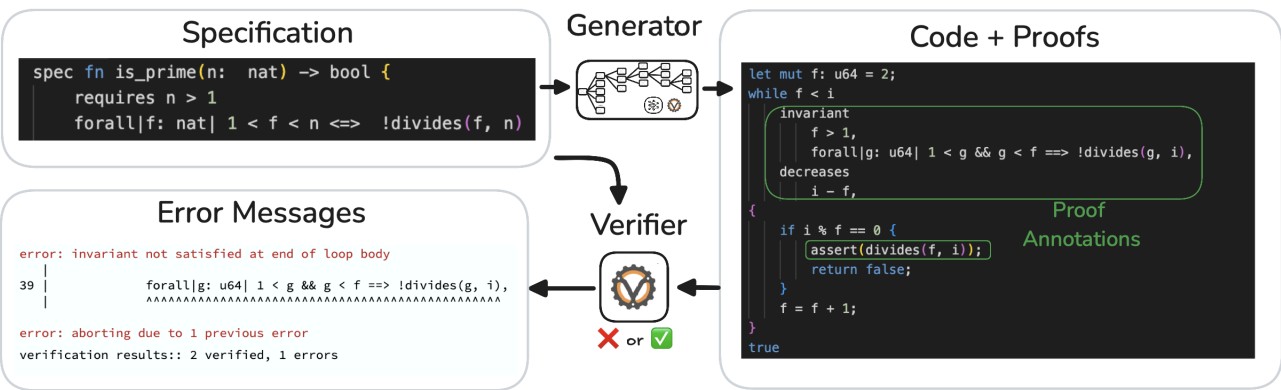

Figure 2: **Example of formally verified code generation.** Given a specification, `AlphaVerus` generates the corresponding code and proof. The verifier checks the proof and provides either verification success or detailed error messages.

### 3.1. Translation

`AlphaVerus` translates programs using a three-stage pipeline consisting of *exploration*, *refinement*, and *critique*. The exploration stage translates source programs into candidate Verus programs. The refinement stage repairs the programs using a novel tree search over program refinements. The critique stage uses a suite of models to discard flawed specifications and implementations that could degrade future iterations. The pipeline iterates, creating a self-reinforcing cycle where verified programs and refinement trajectories improve the models' capabilities, enabling translation of increasingly complex programs. The result is a growing synthetic dataset of progressively more complex and reliable Verus programs. The complete algorithm is listed in Algorithm 1 and visualized in Figure 1.

**Exploration.** Given a source program $x$ (e.g., a Dafny implementation, specification, and proofs), exploration uses a model to generate candidate target (i.e., Verus) programs:

$$\{y_1, \ldots, y_k\} \sim G_{explore}\left(x; D_{x \to y}^{(i)}\right), \quad (1)$$

where $G$ is a generation algorithm (e.g., LLM sampling) that is given the source and a set of (source, target) examples $D_{x \to y}^{(i)}$. Initially, $D_{x \to y}^{(0)}$ has a few hand-written examples.

Any generated (source, verified program) pairs are placed in a candidate set, $C$, that will be passed to the filtering stage. If no candidates verify for source $x$, candidates that are syntactically correct proceed to refinement. Intuitively, this stage serves as initial "exploration", in that it generates a set of candidates that may eventually be refined and filtered into verified programs in the later stages. Unlike other methods of bootstrapping (Zelikman et al., 2022; Lin et al., 2025) that discard anything but correct solutions, we use both syntactically correct programs and fully verified programs for further improvement, expanding the learning signal.

**Refinement with Treefinement.** Having a verifier opens the possibility of refining candidate programs into verified ones by providing detailed feedback, including unverified functions and specific errors like overflows, unsatisfied conditions, and syntactic mistakes (e.g., Figure 2). While human programmers often use such feedback for iterative corrections, naively providing LLMs with incorrect solutions and feedback often fails to produce improvements. Our key insight is that verifier feedback induces an implicit ordering of solutions based on verified functions and error severity. This ordering lets us extend common refinement techniques by framing refinement as a tree search over the space of refined programs, which we call *Treefinement*.

Specifically, the refinement stage takes syntactically correct but unverified candidate translations $\{y_1, \ldots, y_{k'}\}$ and performs a tree search to discover verified programs. Each node in the tree contains an imperfect program and its associated errors, $(y, e(y))$. Nodes are expanded by invoking a refinement model:

$$\{y_1', \ldots, y_k'\} \sim G_{refine}\left(y, e(y); D_{y \to y'}^{(i)}\right), \quad (2)$$

where $D_{y \to y'}^{(i)}$ is a set of (program, error, correct program) examples, initially containing a few hand-written examples.

Given a node scoring function $v(y) \to \mathbb{R}$ that is used to prioritize nodes, we can search over the space of program refinements with a tree search algorithm that selects and expands nodes, such as breadth-first or depth-first search.

We develop a symbolic scoring function based on the number of (un)verified functions, errors, and warnings:

$$s(y) = \frac{n_{\text{ver}}(y) - \alpha n_{\text{err}}(y) - \beta n_{\text{warn}}(y)}{n_{\text{ver}}(y) + n_{\text{unver}}(y)}$$

where $n_{\text{ver}}(y)$ is the number of verified functions in $y$, $n_{\text{err}}(y)$ and $n_{\text{warn}}(y)$ are the counts of errors and warnings from the verifier for the node's program $y$. $\alpha$ and $\beta$ are

hyperparameters controlling the penalties for errors and warnings, respectively. Intuitively, programs that are closer to a verified program have higher scores, with proximity determined by the proportion of verified functions, resolved errors, and resolved warnings. Upon generating a verified program, the program's search trajectory is added to a candidate set $C_\tau$, and the new (source, program) pair to the candidate set $C$ that is passed to the critique stage.

Treefinement extends two kinds of prior methods into a new search over program refinements. First, refining LLM outputs is a common technique (Madaan et al., 2023; Kamoi et al., 2024), but not within a tree search. On the other hand, tree search developed in step-by-step mathematical problem solving involves appending solution steps rather than refining a full program (Wu et al., 2024). Our approach specifically addresses the non-local nature of error fixes.

Although Treefinement can use any tree search algorithm, we use REBASE (REward BAlanced SEarch) (Wu et al., 2024). REBASE allocates an exploration budget by sampling nodes from a distribution determined by the node scores at the current depth, providing an effective balance of exploration and exploitation. The search continues until it finds a verified program or reaches a maximum depth.

**Critique.** Synthesized specifications are the one part of the translation pipeline that lacks formal guarantees, which can result in a mismatch between the intended and actual functionality of generated programs. Furthermore, in a few degenerate cases, there can be a mismatch between the specification's intent and the program's implementation, since Verus has features that can result in trivial programs passing the verifier (e.g., `assume(false)`). These can lead to reward hacking, causing a snowballing effect when used as exemplars in future iterations. Hence, we propose a three-part approach for filtering out such misaligned programs: a *rule-based model*, a *comparison model*, and an *exploit model*.

The rule-based model receives a generated program $y$, and detects if $y$ uses a Verus feature which leads to a trivial program. Since there are a relatively small number of such features, and these features can be detected through string matching, it suffices to use a list of hand-coded filters. This includes checking for `assume(false)`, "`#[verifier::external]`", and trivial preconditions.

The comparison model $f(x, y)$ receives a source input $x$ and a program candidate $y$, and evaluates whether the specifications and algorithms used in the candidate match those from the source in intent and structure. In practice, we prompt a model to generate multiple evaluation sequences and reject an output if at least $r$ sequences indicate rejection.

The exploit model is an adversarial approach that leverages the feedback from Verus. We use a generator prompted to generate simple and often trivial solutions–such as returning an empty array–that satisfy the specifications, i.e., $(y_I, y_P) \sim G_{\text{exploit}}\left(y_S; D_{\text{exploit}}^{(i)}\right)$, where $y_S$ is a generated specification, $y_I, y_P$ is an implementation and proof, and $D_{\text{exploit}}^{(i)}$ contains (specification, implementation+proofs) examples. If such simple solutions pass verification, it indicates that the specification is flawed, and the corresponding translation is discarded. This includes subtle forms of misspecification; for instance, on tasks requiring array manipulation, the specification may omit conditions on the array length, resulting in trivial solutions.

**Self-improvement.** Finally, the newly generated programs and a subset of the error trajectories are added to a pool of data that is used by the translator, refinement, and critique models in the next iteration. In this sense, the models "self-improve" given access to the Verus environment, so long as the generated examples are useful exemplars.

Formally, for exploration, we create a new pool of examples,

$$D_{x \to y}^{(i+1)} = D_{x \to y}^{(i)} \cup \tilde{D}_{x \to y}^{(i+1)}, \qquad (3)$$

where $\tilde{D}_{x \to y}^{(i+1)}$ consists of the (source, program) candidates $C$ that were collected during exploration and refinement, and that additionally pass the critique stage.

For refinement, we create a new pool of examples using the successful trajectories $C_\tau$ collected during refinement. Namely, we keep those trajectories whose final program passes the critique stage, and pair each intermediate program $y$ and its errors with the final program $y'$, i.e.,

$$\tilde{D}_{y \to y'}^{(i+1)} = \{(y, e(y), y') \mid y \text{ is an ancestor of } y'\}, \qquad (4)$$

and set $D_{y \to y'}^{(i+1)} = D_{y \to y'}^{(i)} \cup \tilde{D}_{y \to y'}^{(i+1)}$.

Similarly, for the exploit model, we add (specification, program) exploits that pass the verifier into $D_{\text{exploit}}^{(i+1)}$ to be used by the exploit model in the next iteration. Table 4 summarizes the components, feedback sources, models, and generated synthetic data at each stage.

To use synthetic data as in-context exemplars, we employ a stochastic few-shot sampling approach. Specifically, each time a generator is called, we randomly sample $k$ examples from its respective data pool. This method reduces the computational cost associated with fine-tuning large models and, as shown in our results, enables other models to leverage the data pool to improve their performance without any training. Nevertheless, fine-tuning models and developing learning objectives remain interesting future directions.

**Source domain: Dafny.** As our initial source domain, we consider Dafny–a language that follows a similar paradigm to Verus and has been in use for over a decade, resulting

in a larger set of available data. We use `DafnyBench` (Loughridge et al., 2025), a dataset of 562 Dafny programs.

Translating Dafny programs to Verus presents several challenges due to two major differences: 1. *Language Constructs:* Significant differences exist in supported features, data types, and the design of the underlying verifier, rendering direct translations infeasible. 2. *Proof Requirements:* Verus imposes more rigorous proof obligations, such as overflow checks, making proofs harder to verify.

### 3.2. Downstream Evaluation

After generating high-quality synthetic data in the form of formally verified Verus programs and error-feedback-correction triples, we use the data to enable a model that performs formally verified code generation. Unlike prior work that requires LLMs to fill proof annotations in existing code and specifications (Loughridge et al., 2025; Yang et al., 2024; Chen et al., 2024), we evaluate our models on the more challenging task of generating both the code and the proofs given only the specifications.

We use a two-part approach consisting of *exploration* and *Treefinement*. During exploration, given a specification $y_s$, we generate $k$ candidate programs $\{y^{(1)}, \ldots, y^{(k)}\}$. If any candidate passes verification, we consider the task solved. Otherwise, we initialize Treefinement with the candidates and run it until we obtain a verified solution or reach a maximum number of iterations. This can be seen as a generator that uses the collected data as a source of few-shot exemplars, $(y_I, y_P) \sim G(y_S; D_y, D_{y \to y'})$, which means generating an implementation and proofs using a language model prompted with a subset of the collected verified programs $D_y$ and a test specification $y_S$, followed by Treefinement with the collected refinement examples $D_{y \to y'}$.

## 4. Experimental Setup

**Generators.** We use `LLaMA-3.1-70B` for translation experiments and additionally evaluate `LLaMA-3.1-8B`, `Qwen-32B`, and `GPT-4o` for downstream tasks. The exploration phase uses $k = 256$ samples, while tree search uses breadth $= 32$ and maximum depth $= 8$.

**Translation.** We use `DafnyBench` consisting of 562 programs as our source domain $D_{src}$ for our translation experiments. The exploration model $G_{explore}$ is initialized using a Verus syntax file and 5 examples from the Verus repository.

**Downstream Evaluation.** We evaluate formally verified code generation, where models must generate both an implementation and proof annotations given a specification. We measure Pass@K, where success requires at least one correct solution of the $K$ generated programs.

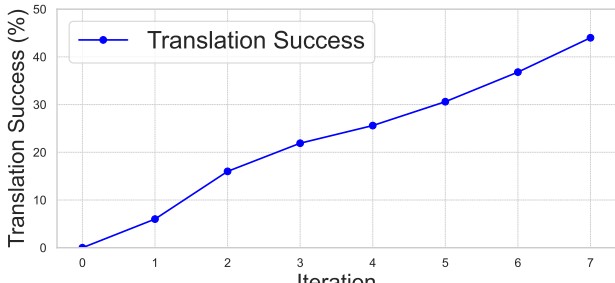

Figure 3: **Programs translated over iterations**. The translation success rate consistently improves over iterations.

**Datasets.** We evaluate on verified versions of the `MBPP` and `HumanEval` datasets. In particular, `MBPP`-verified is sourced from (Yang et al., 2024; Misu et al., 2024) and contains 78 programs from the original MBPP dataset (Austin et al., 2021). `HumanEval`-Verus is sourced from a concurrent open-source effort (The HumanEval-Verus Contributors, 2024) to translate existing `HumanEval` programs to Verus. For brevity, we refer to `HumanEval` and `MBPP` as their respective verified versions throughout this paper.

**Baselines.** Our primary evaluation is performed on verified code generation. Since no existing baselines exist for the task, we use few-shot variants (Listing C) of base models. We tried our best to adapt AutoVerus (Yang et al., 2024) to verified code generation, but due to the complexity of its hand-written prompts, we were not able to achieve non-trivial performance. Hence, we compare `AlphaVerus` on the `MBPP` *proof annotation task* against SAFE++ (Chen et al., 2024) and AutoVerus (Yang et al., 2024).

We refer readers to Appendix A.2 for more details.

## 5. Results and Analysis

**`AlphaVerus` translation success monotonically increases.** Figure 3 shows the number of successful translations over each iteration. We see a steady increase in the number of translations as the iterations increase. The results indicate that `AlphaVerus` learns to translate and generate more complex programs over iterations. Altogether, `AlphaVerus` translates around 45% of DafnyBench into Verus programs that are verified by Verus and aligned according to the critique models. Listings 2, 3, and 4 in the Appendix show example translations.

The exemplars generated during the translation process are collected into the DAFNY2VERUS-COLLECTION, totaling 247 translated programs, 102 error trajectories, and 579 exploit pairs. We use these exemplars for downstream tasks.

**`AlphaVerus` enables verified code generation.** Table 1 shows the verified code generation performance for the `AlphaVerus` model obtained from the final translation

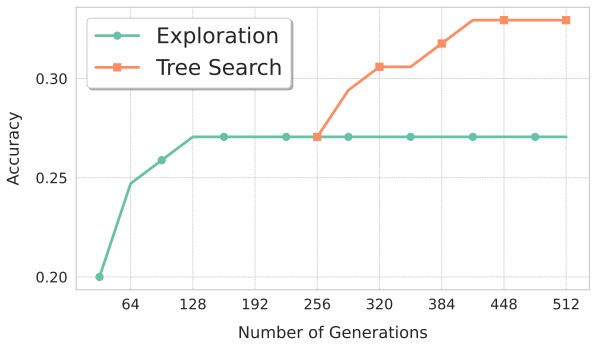

Figure 4: Treefinement vs. exploration (HumanEval). Treefinement leads to a jump in performance that cannot be obtained by additional parallel sampling (exploration).

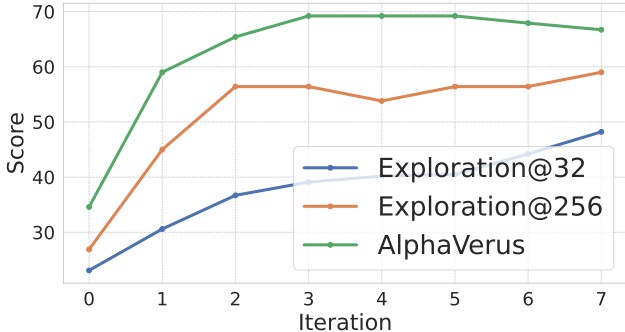

Figure 5: Translation iteration (x-axis) vs. downstream task performance on HumanEval (y-axis). Performance of pass@32 continues to improve, with pass@256 leveling off.

| Method | HumanEval | MBPP |
|---|---|---|
| *Baselines* | | |
| GPT-4o | 27.1% | 35.9% |
| Llama 3.1 70B | 11.8% | 26.9% |
| *Ablations (Treefinement Variants)* | | |
| Single-Turn Linear Self-Refine | 29.4% | 61.5% |
| Multi-Turn Linear Self-Refine | 29.4% | 62.8% |
| Best-First Search | 28.2% | 61.5% |
| **AlphaVerus** (Llama 3.1 70B) | | |
| Exploration | 27.1% | 59.1% |
| + Treefinement (Rebase) | **32.9%** | **65.7%** |

Table 1: **Verified code generation performance** on the `HumanEval` and `MBPP` benchmarks (pass@256).

iteration. `AlphaVerus` leads to a substantial increase over its underlying Llama 3.1 70B model and a prompted GPT-4o model. Moreover, Treefinement leads to an additional increase in performance over the exploration stage. Listings 1, 5, and 6 show example generations. Next, we analyze the impact of the various components in `AlphaVerus`.

**Treefinement leads to a jump in performance.** We evaluate the effectiveness of tree search compared to further scaling the parallel sampling (exploration) budget without refinement. Figure 4 shows the percentage of solved problems versus the generation budget for both approaches. Treefinement leads to a substantial jump in performance over exploration. Notably, exploration plateaus while tree search continues improving as the generation budget is increased.

**Critique is crucial for preventing reward hacking.** Without the critique phase, our analysis of 100 `DafnyBench` examples reveals the model learns to game the verification system by using `assume(false)` statements, leading to

trivially verified but incorrect implementations. We observe a snowballing effect where this behavior spreads across all programs (see Figure 6). While such cases can be disallowed as done by our rule-based critic model, we find more complicated reward hacking instances, such as incomplete specifications and degenerate translations (detailed in Figure 7). The results show the need for our 3-model critique phase for preventing reward hacking.

**Treefinement outperforms linear refinement.** We compare Treefinement against standard refinement that refines linearly, either by performing one step of refinement across multiple parallel branches or performing several steps of refinement across branches. Using equivalent generation budgets, we adjust the breadth and depth parameters accordingly. We also evaluate the best-first search as a baseline. As seen in Table 1, all methods improve upon initial exploration, demonstrating Treefinement's compatibility with various search algorithms, and tree-search based refinement outperforms linear refinement. For the tree search, using REBASE outperforms the best-first search. Also note that the linear refinement variants are special cases of REBASE ($depth = 1$ with large breadth, and $temperature = \infty$).

**AlphaVerus exemplars transfer to other models.** A key advantage of `AlphaVerus` is its ability to transfer learned exemplars without model weight updates. Concretely, we use the exemplars collected during `AlphaVerus`'s translation phase, which used Llama 3.1 70B (i.e., the DAFNY2VERUS-COLLECTION), to enable verified code generation on various models using the same few-shot prompting strategy outlined in §3.2. Table 2 shows successful transfer to both smaller and larger models, yielding significant improvements in verified code generation. Notably, we set a new state-of-the-art on both `HumanEval`, using `GPT-4o` but without finetuning.

**AlphaVerus enables strong proof annotation.** Unlike prior works that focus on the proof annotation task (gen-

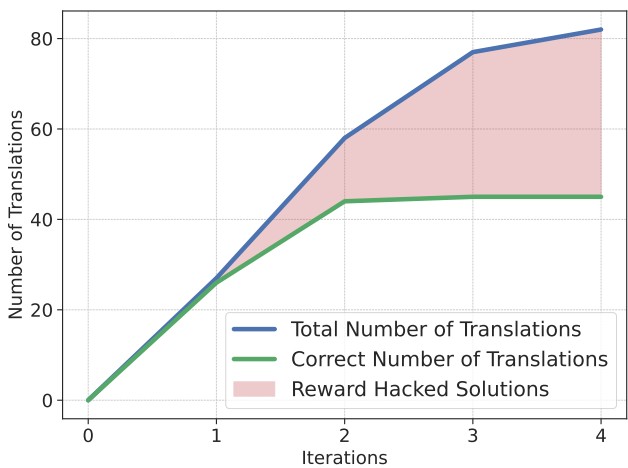

Figure 6: Impact of removing the critique models. Without filtering mechanisms, the model learns to exploit verification by increasingly using `assume(false)` statements. This snowballing effect shows the importance of critique models in preventing reward-hacked solutions.

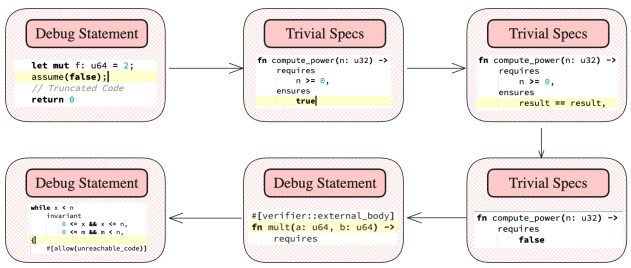

Figure 7: Illustration of reward hacking without the critique models. In particular, the agent first learns to use debug statements and uses them continuously. After fixing, it learns other hacks such as generating trivial specifications or exploring rare debug statements such as allowing infinite loops.

erating proofs given correct code), our method tackles the harder task of formally-verified-code generation, where the model must generate both code and proofs. Despite this, we outperform SAFE (Chen et al., 2024) and AutoVerus (Yang et al., 2024) by 17% and 10% respectively on the proof annotation task (see Table 3). This is notable given that AutoVerus was specifically designed for this task while using significantly more engineering effort and task-specific prompts. Moreover, we achieved these results using just 562 Dafny programs and an open 70B model, compared to SAFE's month-long GPT-4 invocations and training on thousands of programs. Overall, the results point to the effectiveness of AlphaVerus, along with its flexibility and data efficiency. Appendix D.1 contains additional details.

**Other experiments.** We manually inspect translations from each iteration of AlphaVerus to understand what the model learns over iterations. Figure 8 shows the new con-

| | HumanEval |
|---|---|
| Llama 8B - Few Shot | 11.8% |
| + DAFNY2VERUS-COLLECTION | **18.8%** |
| Qwen-32B - Few Shot | 14.1% |
| + DAFNY2VERUS-COLLECTION | **27.1%** |
| GPT-4o - Few Shot | 27.1% |
| + DAFNY2VERUS-COLLECTION | **37.7%** |

Table 2: Transfer of DAFNY2VERUS-COLLECTION to other language models without finetuning. All models show significant improvements over their few-shot variants.

| | MBPP |
|---|---|
| SAFE | 59.0% |
| AutoVerus | 65.4% |
| AlphaVerus | **75.7%** |

Table 3: Comparison of proof annotation generation tasks against baselines on the MBPP dataset.

cepts learned over iterations, showcasing how the model starts by learning how to translate basic syntax, followed by numeric algorithms, and array-related concepts.

Further, Listing 2, 3, 4 show example Dafny-to-Verus translations from AlphaVerus, indicating that AlphaVerus is capable of complex translations often spanning multiple specifications, helper functions, and proof annotations reaching over 100 lines of Verus code. For formally verified code generation, Listing 1 shows an example generated by AlphaVerus; more examples showcasing AlphaVerus's ability to generate multi-function programs, and non-trivial lemmas are shown in Appendix D.2.

Finally, in Figure 9, we show that for lower budgets, using smaller models such as LLaMA-3.1-8B performs better than larger LLaMA-3.1-70B at the same cost. However, LLaMA-3.1-70B achieves higher asymptotic performance, analogous to prior work (Wu et al., 2024).

## 6. Related Work

**Automated Formal Verification.** Automated formal verification has a long-standing history in interactive theorem provers like Coq (Coq Development Team, 2020), Lean (Lean FRO), and Isabelle (Isabelle), with approaches generating step-by-step proofs (Redmon & Sanchez-Stern, 2016; Kaliszyk et al., 2018; Polu & Sutskever, 2020; First et al., 2020; Lu et al., 2023; Li et al., 2024). While successful for mathematical theorems, application to code verification remains limited to existing project theorems (First et al.,

Listing 1: `AlphaVerus` implementation of the unique_sorted function, which removes duplicates from a sorted vector.

```
1   // ----- Function signature and specification (given to model) -----
2   fn unique_sorted(s: Vec<i32>) -> (result: Vec<i32>)
3       requires
4           forall|i: int, j: int| 0 <= i < j < s.len() ==> s[i] <= s[j],
5       ensures
6           forall|i: int, j: int| 0 <= i < j < result.len() ==> result[i] < result[j],
7           forall|i: int| #![auto] 0 <= i < result.len() ==> s@.contains(result[i]),
8           forall|i: int| #![trigger s[i]] 0 <= i < s.len() ==> result@.contains(s[i]),
9   {// ----- Model generation starts here -----
10      let mut result = Vec::new();
11      let mut i: usize = 0;
12      while i < s.len()
13          invariant
14              result.len() <= i,
15              i <= s.len(),
16              forall|j: int| 0 <= j < result.len() ==> s@.contains(result[j as int]),
17              forall|j: int| 0 <= j < i ==> s@[j as int] == s[j as int],
18          {
19              if i == 0 || s[i] != s[i - 1] {
20                  result.push(s[i]);
21              }
22              i += 1;
23          }
24      proof {
25          assert(forall|i: int, j: int| 0 <= i < j < result.len() ==> result[i] < result[j]) by {
26              assert(forall|i: int, j: int| 0 <= i < j < s.len() ==> s@[i] <= s@[j]);
27              assert(result.len() == s.len());
28              assert(result@.len() as int == s@.len() as int);
29              assert(result@.to_multiset() =~= s@.to_multiset());
30          }
31      }
32      result
33  }
```

2023) or simple properties (Lohn & Welleck, 2024). Integrating language models with verification languages such as Dafny (Leino, 2010; Sun et al., 2024a; Loughridge et al., 2025) and F* (Swamy et al., 2016; Chakraborty et al., 2025) face real-world applicability challenges.

**Rust Verification & Concurrent Work.** Verus (Lattuada et al., 2023) brings verification to Rust but suffers from data scarcity. Recent concurrent works address this through GPT-4 pipelines (AutoVerus (Yang et al., 2024), requiring significant human effort) and dataset translation (SAFE++ (Chen et al., 2024), using month-long GPT-4 generation). `AlphaVerus` differs by using open models, 100× less data, and introducing tree-search refinement with critique for verified code generation (implementation + proofs), outperforming their linear refinement strategies.

**Inference-Time Strategies.** Meta-generation strategies (Welleck et al., 2024) boost reasoning via parallel sampling (Wang et al., 2022; Aggarwal et al., 2023; Sun et al., 2024b), tree search (Yao et al., 2024; Wu et al., 2024), and refinement (Welleck et al., 2023; Madaan et al., 2023; Snell et al., 2025). Unlike previous methods that perform stepwise verification (Wu et al., 2024) with a separate reward model, Treefinement combines different verification sources such as scalar values, language feedback and refines complete programs, addressing the non-local nature of error fixes.

**Self-Improvement in LLMs.** Recent work explores improving language models through self-generated data and external feedback (Zelikman et al., 2022; Wang et al., 2025; Hosseini et al., 2024), using expert iteration or rejection sampling strategies. Unlike these sample-and-filter strategies, `AlphaVerus` self-improves using different feedback sources, with data collected from multiple modules.

We refer readers to Appendix B for a more detailed discussion of related work.

## 7. Conclusion

We introduced `AlphaVerus`, a novel self-improving framework for generating formally verified code in mainstream programming languages. By leveraging iterative translation from a higher-resource language (Dafny) to Verus and utilizing verifier feedback through our Exploration, Treefinement, and Critique stages, `AlphaVerus` overcomes the challenges of scarce training data, reward hacking and the complexity of formal proofs. We hope that the methods proposed in our work, such as the critique models and treefinement, may evolve to handle more complex cases of reward hacking and search with LLMs. Our approach operates without human intervention, hand-engineered prompts, or extensive computational resources, yet achieves significant performance improvements on verified versions of the HumanEval and MBPP benchmarks where prior methods fail. We also contribute a new dataset of formally verified Verus programs, providing valuable resources for future research. `AlphaVerus` opens up new avenues for grounding code generation and developing trustworthy AI-assisted programming tools.

## Acknowledgements

We thank Convergent Research and the OpenAI Researcher Access program. We also thank Alex Bai, Jay Bosamiya, Edwin Fernando, Md Rakib Hossain, Jay Lorch, Shan Lu, Natalie Neamtu, Bryan Parno, Amar Shah, Elanor Tang for their contributions to the version of the HumanEval-Verus benchmark we used in our experiments. This work was funded in part by a gift from VMware, the Future Enterprise Security initiative at Carnegie Mellon CyLab (FutureEnterprise@CyLab), and AFRL and DARPA under Agreement FA8750-24-9-1000.

## Impact Statement

This paper presents work whose goal is to advance the field of Machine Learning. There are many potential societal consequences of our work, none which we feel must be specifically highlighted here.

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

# A. Experimental Details

### A.1. Translation

We use `DafnyBench` as our source domain $D_{src}$ for our translation experiments. Starting with 782 programs, we filter to 562 by excluding those that verify without proof annotations. The exploration model $G_{explore}$ is initialized using a Verus syntax file and 5 examples from the Verus repository.

### A.2. Datasets

We evaluate on verified versions of the `MBPP` and `HumanEval` datasets. In particular, `MBPP`-verified is sourced from (Yang et al., 2024; Misu et al., 2024) and contains 78 programs from the original MBPP dataset (Austin et al., 2021). `HumanEval`-Verus is sourced from a concurrent open-source effort (The HumanEval-Verus Contributors, 2024) to translate existing `HumanEval` programs to Verus. Since each task in `HumanEval`-Verus is typically implemented and verified using multiple functions, we split each program into individual provable functions, ensuring that all dependent functions needed are present. Specifically, we split 49 programs into 85 functions and evaluate methods on these 85 functions. We use a snapshot from November 4th, 2024 with commit hash `ddb9ba3`. For brevity, we refer to `HumanEval` and `MBPP` as their respective verified versions throughout this paper.

### A.3. Hyperparameters

We use consistent decoding parameters, with temperature set to $0.7$, top-p set to $1.0$ and max_tokens set to 2048. For the translation step, we generate 256 examples per program in the translation phase. We set breadth and depth to 32 and 8 in the treefinement stage. $\alpha$ is set to $0.1$ and $\beta$ is set to $0.03$ as defined in Equation 3. We set the rebase node sampling temperature to $0.1$. We generate 32 samples for the comparison model and exploit model. We use the same setting in both inference and translation. For stochastic sampling as described in Equation 3.1, we randomly choose $k/2$ examples from the pool of $k$ exemplars. All sampling is done with a batch size of 32. We do not tune hyperparameters and use the conventional settings throughout. We use the 'gpt-4o-2024-08-06' version for `GPT-4o` modal.

### A.4. Contamination Analysis

Despite independent development of `HumanEval` and `MBPP`, we observe significant overlap between these datasets and `DafnyBench` programs. To mitigate contamination in downstream evaluations, we employ GPT-4 for systematic filtering of collected exemplars. Specifically, we prompt GPT-4 with each collected exemplar paired against individual programs from `HumanEval` and `MBPP`, requesting the identification of similar programs. We generate 4 independent evaluations per pair and flag contamination when similarity is detected in more than two evaluations. Flagged examples are excluded from the in-context examples during evaluation of the corresponding program. The prompt used is in Listing C.

Manual analysis confirms this approach significantly outperforms traditional n-gram analysis and aligns well with human assessment of contamination. We recommend future work adopt similar contamination detection methods rather than relying solely on n-gram analysis for program similarity. Notably, existing baseline methods for proof annotations in Verus (Yang et al., 2024; Chen et al., 2024) lack such contamination analysis.

### A.5. Hardware and Software

We use L40S GPUs for inference. We use SgLang for inference (Zheng et al., 2024). We design a scalable and parallel version of the translation and inference stage, where each program is run on a separate node. We release the complete codebase and our DAFNY2VERUS-COLLECTION for reproducibility.

# B. Related Work

**Automated Formal Verification.** Automated formal verification has a long-standing history in interactive theorem provers (Redmon & Sanchez-Stern, 2016; Kaliszyk et al., 2018; Polu & Sutskever, 2020; First et al., 2020; Lu et al., 2023; Li et al., 2024), such as Coq (Coq Development Team, 2020), Lean (Lean FRO), and Isabelle (Isabelle). These approaches typically generate step-by-step proof statements for a given problem, with the theorem prover providing feedback on intermediate steps. While these methods have achieved significant success in proving complex mathematical theorems, their application to formal verification of code is typically limited to theorems from existing projects (e.g., First et al. (2023))

or simple program properties (Lohn & Welleck, 2024) rather than end-to-end verified code generation. An alternative paradigm integrates language models with languages that offload proving to automated reasoning tools (e.g., SMT), including Dafny (Leino, 2010; Sun et al., 2024a; Loughridge et al., 2025) and F* (Swamy et al., 2016; Chakraborty et al., 2025). However, enabling verified code generation in these research languages may have limited applicability to real-world software and workflows.

**Automated Formal Verification in Rust.** In contrast, Verus (Lattuada et al., 2023) offers a verification framework for Rust, a widely adopted programming language. However, unlike in formal theorem proving or long-standing verification languages, there is a substantial lack of data for Verus. Two existing works, released during the development of `AlphaVerus`, attempt to overcome data scarcity. First, AutoVerus (Yang et al., 2024) prompts GPT-4 with a pipeline of hand-engineered prompts tailored to specific errors and programs. This allows for refining some errors but requires human expertise to support new strategies through additional prompts. In contrast, our Treefinement method learns new refinement strategies automatically. Second, the concurrent work SAFE++ (Chen et al., 2024) proposes translating an existing Rust dataset to Verus and training generation and refinement models on the collected data. However, the translation process in Chen et al. (2024) was initialized with over a month of continuous generation from GPT-4. In contrast, `AlphaVerus` relies only on a single openly available model, without an expensive GPT-4 initialization. `AlphaVerus` also incorporates a new tree-search refinement strategy that outperforms the linear strategy used in SAFE++, and a critique phase to ensure the generated specifications are high quality. These innovations contribute to better results, despite our method using open models and 100 times less data. Finally, these two existing works study the simplified task of proof generation, while we study the more general setting of verified code generation: generating the implementation and its proofs.

**Inference-Time Strategies.** Recent studies have shown that increasing inference-time compute can improve performance in reasoning, mathematics, and code generation via meta-generation strategies (Welleck et al., 2024) such as parallel sampling (Wang et al., 2022; Aggarwal et al., 2023; Sun et al., 2024b), tree search (Yao et al., 2024; Wu et al., 2024), and refinement (Welleck et al., 2023; Madaan et al., 2023; Snell et al., 2025). Our Treefinement algorithm can be viewed as a hybrid meta-generator that combines tree search and refinement, following initial parallel sampling (exploration). A variety of tree search methods generate one step of a mathematical solution at a time, with a verifier guiding the search process by assigning a score to the current state (Wu et al., 2024). In contrast, Treefinement uses verifier feedback on the complete solution, modeling tree nodes as full programs and edges as refinement steps. Our strategy addresses the non-local nature of error fixes, and does not need an additional trained scoring model.

Various refinement strategies use external feedback from knowledge bases (Peng et al., 2023; Chern et al., 2023), code interpreters (Chen et al., 2023; Zhang et al., 2023), tool outputs (Gou et al., 2024; Schick et al., 2023), or separately trained reward models (Akyürek et al., 2023). Our Treefinement algorithm uses a diverse set of feedback sources, including scalar and binary values, language feedback, and an exploit model. Moreover, whereas prior methods typically operate in a linear fashion–i.e., starting with an output and repeatedly refining it–our approach structures refinement as a tree search. This allows for prioritizing certain branches of refinement, which we find perform better.

**Self-Improvement in LLMs.** Various algorithms aim to improve a language model using data generated by the model along with an external feedback source (Zelikman et al., 2022; Wang et al., 2025; Hosseini et al., 2024), which is colloquially termed *self-improvement*. Common approaches rely on variants of expert iteration or rejection finetuning (Polu et al., 2022; Zelikman et al., 2022; Yuan et al., 2023; Lin et al., 2025), where multiple solutions are sampled, and an external signal selects the positive ones for model fine-tuning. Our approach, `AlphaVerus`, builds upon these concepts but moves beyond the simple sample-and-filter strategy. Our method additionally uses refinement and tree search to collect data, and the data is collected using multiple modules (e.g., outputs from Treefinement may be used to improve exploration). Additionally, `AlphaVerus` uses various forms of feedback–such as trinary, scalar, language, and verifier outputs–rather than just binary filtering. Conceptually, we can view `AlphaVerus` as a meta-generation algorithm (i.e., a combination of parallel sampling, refinement, and tree search) that improves over time, rather than a model trained on filtered outputs.

## C. Methodology

**Components** Table 4 summarizes the components of our method at different stages, the feedback sources used, the models employed, and the data collected for bootstrapping.

| Stage | Feedback | Model | Data Collected |
|---|---|---|---|
| Exploration | Verifier (errors) | LLM + Parallel Sampling | Verified Translations |
| Treefinement | Verifier (value), Verifier (errors) | LLM + Tree Search + Refinement | Error Fix Triplets, Verified Translations |
| Critique Module | Rules, Trivial Programs, Verifier (binary), Comparison LLM | Regex, String Manipulation, Prompted LLM, Exploit LLM | Exploit Pairs |

Table 4: Different components used in iterative translation in `AlphaVerus`

**Alorithm and Prompts**    We detail the complete algorithm for `AlphaVerus` in Algorithm 1. We list the prompt used for Exploration stage in Listing C, prompt used for Treefinement stage in Listing C, prompt used for exploit and comparison model in Listing C and Listing C, and for inference in Listing C. Unless specified in the prompt, we use user, assistant pairs to simulate few-shot examples.

---

**Verus Code Completion**

Consider the following incomplete Verus code:

```
{program}
```

The code contains the relevant spec functions and the preconditions (`requires`) and postconditions (`ensures`) for the main function. Your goal is to complete the function by adding the necessary procedure, along with proof statements (such as `invariants`, `asserts`, `proof` blocks, etc.) to prove the program.
Only output the new program and not the entire code. You are not allowed to create new functions; however, you can use any functions already defined if they are within the scope.

---

**Translation: Exploration Prompt**

Consider the following dafny code:

```
{program}
```

Your goal is to convert the code to Verus code. Based on the syntax I gave you, convert the code to Verus. Note that you may need to make some datatype-related changes for it to work in Verus. Specifically, use the most appropriate ones from the syntax and code examples provided earlier. However, do not change invariants or specifications (ensures and requires clauses). Make sure to include the use statements, proper start of code using verus!, and empty fn main() as done in the examples.

## Translation Treefinement Prompt

SYSTEM: Here are some examples of fixing verus code based on compiler error message:

```
# Verus Error Fixing Example {i+1}:
## Incorrect Code:
```rust
{incorrect_code}
```
## Error Message:
```
{error_message}
```
## Corrected Code after fixing the errors:
```rust
{corrected_code}
```
```

```
<Other Examples>
```

USER:

Given a Verus program with function signature, preconditions, postconditions, and code, fix the errors present in the code. Effectively return the complete verys program by fixing all proof statements or adjusting the code, such that the code compiles correctly. Do no modify function signatures requires, ensures or specs. Repeat: Do not ever modify those lines in ensures clause, requires clause, function signatures. Just edit the proof. **Only in case of overflow errors**, you can make reasonable relaxations on the size of the input variables. For instance, considering the input length of array to be any value less than 10 is not reasonable. Similarly for integer inputs, considering them to be small numbers is not reasonable. Choose bigger bounds for relaxation. You can also use spec functions, to estimate the max value, and impose a condition accordingly. For instance, if error is integer overflow while doing multiplication, you can add requires statement such as:

```
forall|k: int| 0 <= k < nums.len() ==> (0 <= #[trigger] nums[k] * #[trigger]
    nums[k] < i32::MAX)
```

However, absolutely no other changes to precondition and postcondition are permitted! Below is the program::

```
{program}
```

The program has following error message:

```
{error_messsage}
```

Solution Format:

```
[Thoughts on Error Message]
[Thoughts on Error Resolution]
[Thoughts on Corner Cases, such as Overflow etc.]
```rust
[Complete Code]
```
```

## Translation: Exploit Model Prompt

You are a Verus exploit agent that finds trivial solutions for incomplete and inaccurate preconditions and postconditions. Your goal is to complete the code by proposing trivial solutions that pass all verification conditions. Here are some examples:
## Input Problem:

```
use vstd::prelude::*;

verus! {

// Define a function to calculate the nth power of 2
fn power(n: u32) -> (result: u32)
    [Code Truncated]
}

// Define the function ComputePower to calculate 2^n for a given n
fn compute_power(n: u32) -> (result: u32)
    requires
        n >= 0,
        n <= 10000, // arbitrary bound, verus can't handle infinite recursion
    ensures
        result == result,
{
```

## Trivial Solution:

```
    let mut result: u32 = 1;
    let mut x: u32 = 0;
    // invariant: 0 <= x <= n, and result == Power(x)
    while x!= n
        invariant
            0 <= x && x <= n,
            result == result, // result == Power(x),
    {
        x += 1;
        result = result.wrapping_add(result);
    }
    result
}

// Main function, empty for now
fn main() {}

} // verus!
```

<Other Examples>

Charactersitics of a trivial solution:
1. Usually 1-5 lines of code
2. Does not use any complex data structures
3. Usually returns constant values, that passes all test cases.
Your task is to provide only the trivially completed code, given a new program. Only output the new program and not the entire code.

---

**Translation: Comparison Model Prompt**

Consider the following function:

```rust
{rust_code}
```

and

```dafny
{dafny_code}
```

Consider the preconditions and postconditions of the various functions in the two programs along with the spec functions if present, that need to be proven.

## Are the preconditions and postconditions from both the programs same? Note, since they are from different programming languages, minor changes are to be ignored. Minor changes include, adding extra preconditions to limit size of input in rust code, so as to ensure overflows are not encountered, or reformulating implication statements. Such changes are not to be considered, and the answer should be yes, if they are same. Further, preconditions on size of input is reasonable, if there is a possibility of overflow. For instance, for computing fibonacci numbers, using something like n¡=47 is reasonable, and answer should be yes. However, using n¡=5 would be incorrect, and answer should be no. Remember, you have to focus on ensures and requires clause of the main function as postconditions and preconditions respectively.

Follow the following format:

[What Preconditions and Postconditions of Program 1]

[What Preconditions and Postconditions of Program 2]

[Step by Step Thoughts on comparison between conditions. Minor differences can be ignored]

Finally, answer in format to the question if they are same: Therefore, both programs are same /not_same, the final answer is Yes/No.

---

**Contamination Check Prompt**

Consider the following set of program database:

```json
<in context exemplars>
```

## Task: Your task is to find the program that is same or very similar ($\geq 50\%$) to this program:

```
{Program from dataset}
```

You should start the solution, by first thinking which programs would be closest and why. Then, you should output the json, containing the same keys as above: prog_num, program_text. It is possible that none of the programs is closest, or even similar. In that case return empty json object.

---

## D. Results

### D.1. Reward Hacking

Next, we analyze the quality of translations without the critique phase. Figure 6 shows the effect of removing the critique models and continuing the self-improvement process on 100 examples from `DafnyBench`. Without the critique phase, the model is able to translate a large fraction of programs, but it is primarily because of learning to use `assume(false)` which renders any implementation trivially verified. This is primarily used by human developers to debug their proofs. However, here `AlphaVerus` figures out how to game the system by generating trivial proofs.

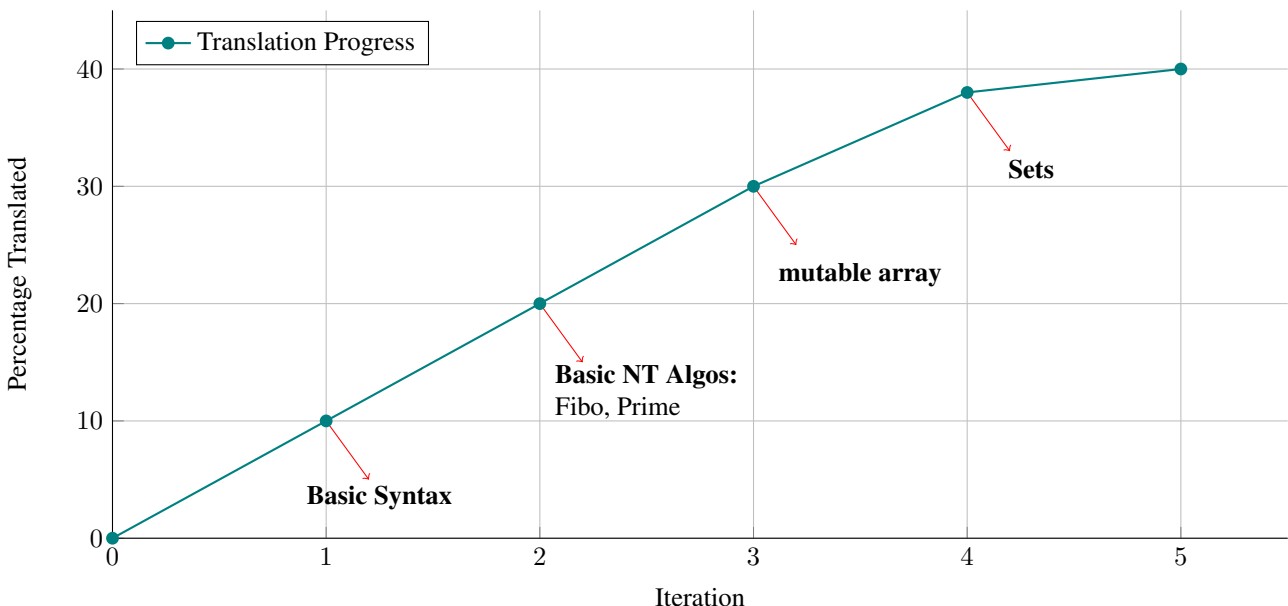

Figure 8: **Translation Progress by Concept**. The graph shows the incremental progress in translation capabilities as different programming concepts are mastered.

There is also a snowballing effect, where initially the model generates a single program with `assume(false)`, then learns to use it in all programs. This is evident from the leveling off of correct translations in the figure. While an obvious way is to disallow such statements (as done by our rule-based verifier), we see even more complicated cases of reward hacking, such as leaving small gaps in translated specifications or even generating degenerate translations, as illustrated in Figure 7. We conclude that the critique phase is critical for filtering out misaligned programs and preventing reward hacking.

**AlphaVerus enables strong proof annotation.** Unlike our work which evaluates methods on the difficult task of formally-verified-code generation that requires generating both code and proof, concurrent work on Verus evaluates on the task of proof annotation: generating proofs given correct code. This is a simpler task since the code is already known to be correct. We compare against SAFE (Chen et al., 2024) using their reported results with DSCoder-33B at Pass@110, as their implementation is not publicly available. We also evaluate against AutoVerus (Yang et al., 2024) using their default configuration with a 70B model.

As shown in Table 3, `AlphaVerus` outperforms SAFE by 17% and AutoVerus by 10%. This is notable since `AlphaVerus` was not designed for the proof annotations task, while AutoVerus has correction prompts specifically engineered for the task. Their engineering also results in reduced generalizability; for instance, AutoVerus cannot be evaluated on `HumanEval` as it doesn't support multi-function programs. Second, SAFE used over a month of GPT-4o invocations and thousands of programs, contrasting with our use of 562 Dafny programs and an openly available 70B model.

**AlphaVerus learns new concepts over iterations.** Next, our goal is to understand what the model learns over iterations that improves its ability to translate more complex programs and improve downstream performance. We manually inspect translations from each iteration of `AlphaVerus` in an attempt to qualitatively characterize the kinds of programs that the system gradually learns to translate. Figure 8 depicts the new concepts that we identified across iterations, starting with the ability to translate basic syntax, then basic numeric algorithms, and then the ability to work with mutable arrays and sets.

**Cost-optimal model for inference.** Next, we compare the performance of different models as we increase the inference cost. We compare `LLaMA-3.1-8B` and `LLaMA-3.1-70B`, using a cost ratio of 1:8 based on current API pricing. That is, generating 8 outputs with `LLaMA-3.1-8B` has the same cost as generating 1 output with `LLaMA-3.1-70B`. We show the accuracy of each model as a function of cost in Figure 9. `LLaMA-3.1-8B` achieves faster initial gains, reaching an accuracy of 0.55 with 128 units of cost, while `LLaMA-3.1-70B` requires about 4 times more cost to reach similar performance. In other words, for cost-constrained scenarios, it is preferable to use the smaller model with more samples, but

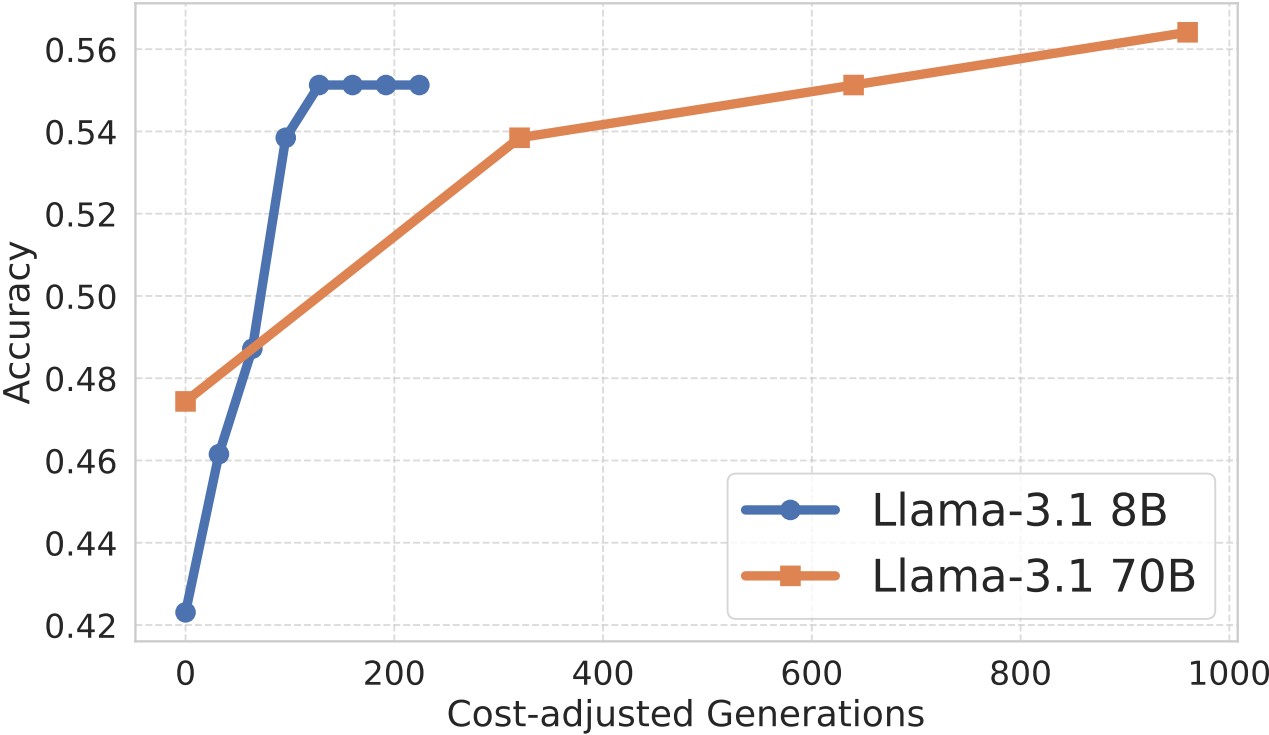

Figure 9: Performance scaling of `LLaMA-3.1-8B` and `LLaMA-3.1-70B` with cost. `LLaMA-3.1-8B` shows better cost efficiency at lower compute budgets, while `LLaMA-3.1-70B` shows higher asymptotic performance.

the larger model has better asymptotic performance. Our findings echo those of Wu et al. (2024) and Snell et al. (2025).

### D.2. Qualitative Analysis

Listing 2, Listing 3, and Listing 4 show example Dafny-to-Verus translations from `AlphaVerus`, indicating that `AlphaVerus` is capable of complex translations. In particular, the translations can involve multiple specifications, helper functions, and proof annotations, and individually reach up to 100 lines of Verus code. For formally verified code generation, Listing 1 shows a generated implementation and proofs for a function that removes duplicates from a sorted vector while maintaining its sorted order. The model is given the function signature and the specification (the `requires` and `ensures` clauses). `AlphaVerus` generates a Rust implementation and proof annotations (e.g., `invariant`, `proof`, `assert` statements) that pass the verifier. Listing 5 shows a multi-function example, in which a helper function `is_prime` is followed by a `largest_prime_factor` function. In addition to function implementations, Listing 6 shows `AlphaVerus` completing a nontrivial lemma.

Listing 2, Listing 3, Listing 4 show example Dafny-to-Verus translation from `AlphaVerus`.

Listing 5, Listing 6 show example formally verified code generations from `AlphaVerus`.

## E. Limitations and Future Work

Our results show promising performance and potential for scaling the methods proposed in this work. In particular, we achieved state-of-the-art performance without any fine-tuning. However, moving forward, exploring fine-tuning models and RL algorithms guided by verifier feedback would be a promising and important direction. Further, while our analysis shows the critique module being effective in preventing reward hacking, we believe investigating better methods for filtering misaligned specifications remains an important future direction. Finally, we believe that treefinement can be extended to other domains, and better search strategies can be developed for even better performance in this important task of verified code generation.

Listing 2: Translation example.

```
1  // ---- Dafny program (input to AlphaVerus) ----
2  function R(n: nat): nat {
3      if n == 0 then 0 else if R(n-1) > n then R(n-1) - n else R(n-1) + n
4  }
5
6  method calcR(n: nat) returns (r: nat)
7      ensures r == R(n)
8  {
9      r := 0;
10     var i := 0;
11     while i < n
12         decreases n-i
13         invariant 0 <= i <= n
14         invariant r == R(i)
15     {
16         i := i + 1;
17         if r  > i {
18             r := r - i;
19         }
20         else {
21             r := r + i;
22         }
23     }
24 }
```

```
1  // ---- Verus program (all model-generated) ----
2  use vstd::prelude::*;
3
4  verus! {
5
6  fn calc_r(n: usize) -> (r: usize)
7      requires
8          n < 100000, // added relaxation to prevent overflow
9          n * (n + 1) / 2 < usize::MAX, // added check to prevent overflow
10     ensures
11         r == n * (n + 1) / 2,
12 {
13     let mut r: u128 = 0;
14     let mut i = 0;
15     while i < n
16         invariant
17             0 <= i && i <= n,
18             true,
19         decreases
20             n - i,
21     {
22         i = i + 1;
23     }
24     assert(i == n);
25     let temp: u128 = n as u128 * (n as u128 + 1) / 2;
26     assert(temp <= usize::MAX as u128);
27     let r: usize = temp as usize;
28     r
29 }
30
31 fn main() {}
32
33 } // verus!
```

Listing 3: Translation example.

```
1   // ---- Dafny program (input to AlphaVerus) ----
2   method SearchRecursive( a: seq<int>, i: int, j: int, x: int ) returns (k: int)
3       decreases j-i;
4       requires 0 <= i <= j <= |a|;
5       ensures i <= k < j || k == -1;
6       ensures k != -1 ==> a[k] == x;
7       ensures k != -1 ==> forall r | k < r < j :: a[r] != x;
8       ensures k == -1 ==> forall r | i <= r < j :: a[r] != x;
9   {
10      if j == i
11      {
12          k := -1;
13          return;
14      }
15      if a[j-1] == x
16      {
17          k := j-1;
18          return;
19      }
20      else
21      {
22          k := SearchRecursive(a, i, j-1, x);
23      }
24  }
25
26  method SearchLoop( a: seq<int>, i: int, j: int, x: int ) returns (k: int)
27      requires 0 <= i <= j <= |a|;
28      ensures i <= k < j || k == -1;
29      ensures k != -1 ==> a[k] == x;
30      ensures k != -1 ==> forall r | k < r < j :: a[r] != x;
31      ensures k == -1 ==> forall r | i <= r < j :: a[r] != x;
32  {
33      if i == j
34      {
35          return -1;
36      }
37
38      var t := j;
39      while t > i
40          decreases t;
41          invariant forall p | t <= p < j :: a[p] != x;
42      {
43          if a[t-1] == x
44          {
45              k := t-1;
46              return;
47          }
48          else
49          {
50              t := t - 1;
51          }
52      }
53      k := -1;
54  }
```

```
1   // ---- Verus program (all model-generated) ----
2   use vstd::prelude::*;
3
4   verus! {
5
6   /// Searches for an element in a sequence using recursion.
7   fn search_recursive(a: &Vec<i32>, i: usize, j: usize, x: i32) -> (k: i32)
8       requires
9           0 <= i && i <= j && j <= a.len(),
10          j - i < 100000, // added precondition
11          j <= i32::MAX as usize, // added precondition
12      ensures
13          i <= k && k < j || k == -1,
14          k != -1 ==> a[k as int] == x,
15          k != -1 ==> forall|r: nat| k < r && r < j ==> a[r as int] != x,
16          k == -1 ==> forall|r: nat| i <= r && r < j ==> a[r as int] != x,
17      decreases
18          j - i,
19  {
20      if j == i {
21          -1
22      } else if a[j - 1] == x {
23          assert(j - 1 <= i32::MAX as usize);
24          if j - 1 <= i32::MAX as usize {
25              (j - 1) as i32
26          } else {
27              -1
28          }
29      } else {
30          search_recursive(a, i, j - 1, x)
31      }
32  }
33
34  /// Searches for an element in a sequence using a loop.
35  fn search_loop(a: &Vec<i32>, i: usize, j: usize, x: i32) -> (k: i32)
36      requires
37          0 <= i && i <= j && j <= a.len(),
38          j - i < 100000, // added precondition
39          j <= i32::MAX as usize, // added precondition
40      ensures
41          i <= k && k < j || k == -1,
42          k != -1 ==> a[k as int] == x,
43          k != -1 ==> forall|r: nat| k < r && r < j ==> a[r as int] != x,
44          k == -1 ==> forall|r: nat| i <= r && r < j ==> a[r as int] != x,
45  {
46      if i == j {
47          return -1;
48      }
49      let mut t: usize = j;
50      while t > i
51          invariant
52              0 <= i <= t && t <= j,
53              t <= a.len(),
54              forall|p: nat| t <= p && p < j ==> a[p as int] != x,
55              t <= i32::MAX as usize, // added invariant
56          decreases
57              t,
58      {
59          assert(t <= a.len());
60          if a[t - 1] == x {
61              if t - 1 <= i32::MAX as usize {
62                  return (t - 1) as i32;
63              } else {
64                  return -1;
65              }
66          } else {
67              assert(t > i);
68              t = t - 1;
69          }
70      }
71      -1
72  }
73
74  fn main() {}
75
76  } // verus!
```

Listing 4: Translation example. We collapse some whitespace and lines with brackets due to space.

```
1   // ---- Dafny program (input to AlphaVerus) ----
2   module OneSpec {
3       datatype Variables = Variables(value: int)
4
5       predicate Init(v: Variables) { v.value == 0 }
6
7       predicate IncrementOp(v: Variables, v': Variables) { && v'.value == v.value + 1 }
8
9       predicate DecrementOp(v: Variables, v': Variables) { && v'.value == v.value - 1 }
10
11      datatype Step =
12          | IncrementStep()
13          | DecrementStep()
14
15      predicate NextStep(v: Variables, v': Variables, step: Step)
16      {
17          match step
18              case IncrementStep() => IncrementOp(v, v')
19              case DecrementStep() => DecrementOp(v, v')
20      }
21
22      predicate Next(v: Variables, v': Variables)
23      { exists step :: NextStep(v, v', step) }
24  }
25
26  module OneProtocol {
27      datatype Variables = Variables(value: int)
28
29      predicate Init(v: Variables)  { v.value == 0 }
30
31      predicate IncrementOp(v: Variables, v': Variables) { && v'.value == v.value - 1 }
32
33      predicate DecrementOp(v: Variables, v': Variables) { && v'.value == v.value + 1 }
34
35      datatype Step =
36          | IncrementStep()
37          | DecrementStep()
38
39      predicate NextStep(v: Variables, v': Variables, step: Step)
40      {
41          match step
42              case IncrementStep() => IncrementOp(v, v')
43              case DecrementStep() => DecrementOp(v, v')
44      }
45
46      predicate Next(v: Variables, v': Variables)
47      { exists step :: NextStep(v, v', step) }
48  }
49
50  module RefinementProof {
51      import OneSpec
52      import opened OneProtocol
53
54      function Abstraction(v: Variables) : OneSpec.Variables {
55          OneSpec.Variables(v.value)
56      }
57
58      lemma RefinementInit(v: Variables)
59          requires Init(v)
60          ensures OneSpec.Init(Abstraction(v)) { }
61
62      lemma RefinementNext(v: Variables, v': Variables)
63          requires Next(v, v')
64          ensures OneSpec.Next(Abstraction(v), Abstraction(v'))
65      {
66          var step :| NextStep(v, v', step);
67          match step {
68              case IncrementStep() => {
69                  assert OneSpec.NextStep(Abstraction(v), Abstraction(v'), OneSpec.DecrementStep());
70              }
71              case DecrementStep() => {
72                  assert OneSpec.NextStep(Abstraction(v), Abstraction(v'), OneSpec.IncrementStep());
73              }
74          }
75      }
76  }
```

```
1   // ---- Verus program (all model-generated) ----
2   #[allow(unused_imports)]
3   use builtin::*;
4   #[allow(unused_imports)]
5   use builtin_macros::*;
6
7   verus! {
8   struct Variables {
9       value: int,
10  }
11  spec fn init(v: Variables) -> bool { v.value == 0 }
12  spec fn increment_op(v: Variables, v_prime: Variables) -> bool { v_prime.value == v.value + 1 }
13  spec fn decrement_op(v: Variables, v_prime: Variables) -> bool { v_prime.value == v.value - 1 }
14
15  enum Step {
16      Increment,
17      Decrement,
18  }
19  spec fn next_step(v: Variables, v_prime: Variables, step: Step) -> bool {
20      match step {
21          Step::Increment => increment_op(v, v_prime),
22          Step::Decrement => decrement_op(v, v_prime),
23      }
24  }
25  spec fn next(v: Variables, v_prime: Variables) -> bool {
26      exists|step: Step| next_step(v, v_prime, step)
27  }
28
29  struct VariablesProtocol { value: int, }
30
31  spec fn init_protocol(v: VariablesProtocol) -> bool { v.value == 0 }
32  spec fn increment_op_protocol(v: VariablesProtocol, v_prime: VariablesProtocol) -> bool { v_prime.value == v.value -
        1 }
33  spec fn decrement_op_protocol(v: VariablesProtocol, v_prime: VariablesProtocol) -> bool { v_prime.value == v.value +
        1 }
34
35  enum StepProtocol {
36      Increment,
37      Decrement,
38  }
39  spec fn next_step_protocol(v: VariablesProtocol, v_prime: VariablesProtocol, step: StepProtocol) -> bool {
40      match step {
41          StepProtocol::Increment => increment_op_protocol(v, v_prime),
42          StepProtocol::Decrement => decrement_op_protocol(v, v_prime),
43      }
44  }
45  spec fn next_protocol(v: VariablesProtocol, v_prime: VariablesProtocol) -> bool {
46      exists|step: StepProtocol| next_step_protocol(v, v_prime, step)
47  }
48  spec fn abstraction(v: VariablesProtocol) -> Variables {
49      Variables { value: v.value }
50  }
51
52  proof fn refinement_init(v: VariablesProtocol)
53      requires
54          init_protocol(v),
55      ensures
56          init(abstraction(v)),
57  { assert(true); }
58
59  proof fn refinement_next(v: VariablesProtocol, v_prime: VariablesProtocol)
60      requires
61          next_protocol(v, v_prime),
62      ensures
63          next(abstraction(v), abstraction(v_prime)),
64  {
65      assert(abstraction(v_prime).value == abstraction(v).value + 1 || abstraction(v_prime).value == abstraction(v).
          value - 1);
66      assert(exists|step: Step| next_step(abstraction(v), abstraction(v_prime), step)) by {
67          if abstraction(v_prime).value == abstraction(v).value + 1 {
68              assert(next_step(abstraction(v), abstraction(v_prime), Step::Increment));
69          } else {
70              assert(next_step(abstraction(v), abstraction(v_prime), Step::Decrement));
71          }
72      }
73      assert(next(abstraction(v), abstraction(v_prime)));
74  }
75
76  fn main() {}
77  } // verus!
```

Listing 5: Multi-function example. First, we have AlphaVerus complete the helper function it_prime given its specification and preceding code. Then we have AlphaVerus complete largest_prime_factor.

```
1   spec fn spec_prime_helper(num: int, limit: int) -> bool {
2       forall|j: int| 2 <= j < limit ==> (#[trigger] (num % j)) != 0
3   }
4
5   spec fn spec_prime(num: int) -> bool {
6       spec_prime_helper(num, num)
7   }
8
9   fn is_prime(num: u32) -> (result: bool)
10      requires
11          num >= 2,
12      ensures
13          result <==> spec_prime(num as int),
14  { // ------- Model generation 1 starts here ---------
15      let mut i = 2;
16      let mut result = true;
17      while i < num
18          invariant
19              2 <= i <= num,
20              result <==> spec_prime_helper(num as int, i as int),
21      {
22          if num % i == 0 {
23              result = false;
24          }
25          i += 1;
26      }
27      result
28  }
29
30  fn largest_prime_factor(n: u32) -> (largest: u32)
31      requires
32          n >= 2,
33      ensures
34          1 <= largest <= n,
35          spec_prime(largest as int),
36          n % largest == 0,
37          forall|p| 0 <= p < n && spec_prime(p) && n as int % p == 0 ==> p <= largest,
38  { // ------- Model generation 2 starts here ---------
39      let mut largest = 1;
40      let mut i = 2;
41      while i < n
42          invariant
43              2 <= i <= n,
44              1 <= largest <= i,
45              spec_prime(largest as int),
46              i <= largest ==> spec_prime(i as int),
47              largest <= n,
48              n % largest == 0,
49              forall|p| 0 <= p < i && spec_prime(p) && n as int % p == 0 ==> p <= largest,
50      {
51          if is_prime(i) && n % i == 0 {
52              largest = i;
53          }
54          i += 1;
55      }
56      assert(1 <= largest && largest <= n);
57      assert(spec_prime(largest as int));
58      assert(n % largest == 0);
59      assert(forall|p| 0 <= p < n && spec_prime(p) && n as int % p == 0 ==> p <= largest);
60      largest
61  }
```

Listing 6: Lemma example. `AlphaVerus` completes the lemma `lemma_step_subrange`.

```
proof fn lemma_step_subrange(substring: Seq<char>, string: Seq<char>)
    requires
        substring.len() > 0,
        string.len() >= substring.len(),
    ensures
        (substring[0] == string.subrange(0, substring.len() as int)[0] && (substring.skip
            (1)
            =~= string.skip(1).subrange(0, substring.skip(1).len() as int))) ==> (
                substring
            =~= string.subrange(0, substring.len() as int)),
    decreases substring.len(),
{ // -------- Model generation starts here --------
    let m = substring.len();
    let n = string.len();
    assert(m > 0);
    assert(m <= n);
    let h = string.subrange(0, m as int);
    assert(h.len() == m);
    assert(substring.len() == h.len());
    if substring[0] == h[0] {
        let k = substring.skip(1).len();
        assert(k == h.skip(1).len());
        let g = string.skip(1).subrange(0, k as int);
        assert(g.len() == k);
        assert(k == substring.len() - 1);
        assert(k == h.len() - 1);
        if m > 1 {
            if substring.skip(1) =~= g {
                assert(substring[0] == h[0]);
                assert(substring.len() == g.len() + 1);
                assert(substring.len() == h.len());
                assert(forall|i: nat| 0 <= i && i < m ==> substring[i as int] == h[i as
                    int]) by {
                    assert(substring[0] == h[0]);
                    assert(substring.skip(1) =~= g);
                    assert(forall|i: nat| 1 <= i && i < m ==> substring[i as int] == g[i
                        as int - 1]);
                    assert(substring =~= h);
                }
                assert(substring =~= h);
            } else {
                assert(!(substring =~= h));
            }
        } else {
            assert(substring =~= h);
        }
    } else {
        assert(!(substring =~= h));
    }
}
```

---

**Algorithm 1** Iterative Translation and Refinement

---

**Input:** Source programs $D_{\text{src}}$, initial data $D_{x \to y}^{(0)}, D_{y \to y'}^{(0)}, D_{\text{exploit}}^{(0)}$
**Output:** Verified target programs $D_{\text{tgt}}$
Initialize $i \leftarrow 0$.
**while** not converged **do**

    **(I) Candidate Generation & Verification:**

      $C \leftarrow \emptyset$: verified pairs;    $S_{\text{unverified}} \leftarrow \emptyset$: (x, y) pairs for refinement
      **foreach** $x \in D_{\text{src}}$ **do**
          Generate candidate translations $\{y_j\} \sim G_{\text{explore}}(x; D_{x \to y}^{(i)})$
          $C_x \leftarrow \emptyset$: verified pairs for this $x$;    $S_x \leftarrow \emptyset$: syntactically correct, unverified candidates for this $x$
          **foreach** $y_j$ **do**
              **if** $y_j$ passes verification **then**
                 $C_x \leftarrow C_x \cup \{(x, y_j)\}$
              **else if** $y_j$ is syntactically correct **then**
                 $S_x \leftarrow S_x \cup \{y_j\}$
        $C \leftarrow C \cup C_x$
        **if** $C_x = \emptyset$ **then**
           $S_{\text{unverified}} \leftarrow S_{\text{unverified}} \cup \{(x, y) | y \in S_x\}$

    **(II) Refinement via Treefinement Search:**

      **foreach** $(x, y) \in S_{\text{unverified}}$ **do**
        Initialize a refinement tree with root node $(y, e(y))$
        **while** max iterations not reached **do**
          Select node $(y', e(y'))$ by REBASE scoring
          Generate refinements $\{y'_k\} \sim G_{\text{refine}}(y', e(y'); D_{y \to y'}^{(i)})$
          **foreach** $y'_k$ **do**
             **if** $y'_k$ passes verification **then**
               $C \leftarrow C \cup \{(x, y'_k)\}$; record trajectory in $C_\tau$
               **break** (stop refining this candidate)
             **else**
               Add $(y'_k, e(y'_k))$ as a child node to the refinement tree

    **(III) Filtering and Data Update:**

      Initialize $D_{\text{exploit}}^{(i+1)} \leftarrow D_{\text{exploit}}^{(i)}$
      **foreach** $(x, y) \in C$ **do**
        **if** critic rejects $y$ or $f(x, y) = $ False or $z \sim G_{\text{exploit}}(s_y; D_{\text{exploit}}^{(i)})$ finds exploit **then**
          Discard $y$
          **if** exploit $z$ is found **then**
            $D_{\text{exploit}}^{(i+1)} \leftarrow D_{\text{exploit}}^{(i+1)} \cup \{(s_y, z)\}$

    Update $D_{x \to y}^{(i+1)} \leftarrow D_{x \to y}^{(i)} \cup C$
    Update $D_{y \to y'}^{(i+1)} \leftarrow D_{y \to y'}^{(i)} \cup \{(y, e(y), y') | (x, y') \in C_\tau\}$
    $i \leftarrow i + 1$
**return** $D_{\text{tgt}} \leftarrow \{y \mid (x, y) \in D_{x \to y}^{(i)}\}$

---

