# OpenReview forum: "AlphaVerus: Bootstrapping Formally Verified Code Generation through Self-Improving Translation and Treefinement"
_ICML.cc/2025/Conference — ICML 2025 poster_

### Official Review · Reviewer_Bjqi · 2025-03-13

**Overall Recommendation:** 4

**Summary:**

The paper proposes a holistic approach for the LLM-based generation of verified code. First, the paper proposes an approach to address the scarcity of training/sample data in many real-world programming languages that could serve as code generation targets. To this end, the paper proposes an LLM-based technique that allows the translation of samples from "data-rich" languages to "data-poor" languages (e.g. from Dafny for which larger benchmark datasets exist to Rust). The paper evaluates the proposed translation pipeline (incl. ablation studies) and demonstrates the applicability of the resulting data set to verified code generation.

## update after rebuttal
I appreciate the authors' response which lifted all open questions on my side.
I keep my score and am in favour of accepting the paper.

**Claims And Evidence:**

The claims of the paper are well supported by empirical evidence.

**Essential References Not Discussed:**

no

**Experimental Designs Or Analyses:**

I checked the experimental results presented in the paper and appendix.
In my view, the paper does a good job at grounding its claims with experimental results.
Unfortunately, there is no baseline to which a direct comparison would be possible.
Nonetheless, the authors put significant effort into comparing to related approaches where possible which is laudable.

**Methods And Evaluation Criteria:**

The paper uses appropriate methods for its objectives.
While the "Critique" step of the translation pipeline does not provide any formal guarantees, the paper specifically discusses this limitation and the approach is nonetheless a significant step forward in correct translation.

**Other Comments Or Suggestions:**

Concerning Algorithm 1:
- It is not clear from the algorithm that $D_{\text{exploit}}$ is used in the if statement.
  If possible, it would be better to also adopt the $\dots \sim G_{\text{exploit}}(\dots)$ notation here
- $D_{\text{exploit}}^{(i+1)}$ seems to miss an initialization with $D_{\text{exploit}}^{(i)}$
- The algorithm looks like $S$ from step (I) is always used -- independently of whether a correct sample is found.
  This is confusing because on page 4 (at least superficially) it sounds like $S$ is only used if no correct sample was found.
  ("If no candidates verify for source $x$, candidates that are syntactically correct proceed to refinement")

**Other Strengths And Weaknesses:**

I appreciate the effort the authors put into the filtering approach.
The experimental results clearly show that this drastically increases the quality of translations, which is great!

However, I have to note that I am not convinced that the rule-based approach can filter out all cases where a specification becomes trivially verifiable. For example `assume(false)` could be rephrased into something like `assume(0!=1)` (or even worse something depending on variables in the program but always evaluating to false).

**Questions For Authors:**

none

**Relation To Broader Scientific Literature:**

The paper indeed addresses an important problem:
For many programming languages that would be natural targets for LLM-based verified code generation or specification generation, there exists relatively little data that can be used for training, fine-tuning or prompting.

LLM-based verification has also been attempted for other real world programming languages such as C with ACSL specifications [arxiv23,FASE24,CAV24] or Java with JML specifications [AI4MATH24,ISOLA24,GPCE24]. In many cases, the availability of benchmarks is an issue. Consequently, AlphaVerus is a very welcome contribution.

Sidenote: I do *not* expect you to cite all these publications, this is just meant to illustrate the wider landscape and emphasize that scarcity of data is a real problem.

[arxiv23] https://arxiv.org/pdf/2311.07948
[FASE24] https://link.springer.com/chapter/10.1007/978-3-031-57259-3_13
[CAV24] https://link.springer.com/chapter/10.1007/978-3-031-65630-9_16
[AI4MATH24] https://openreview.net/forum?id=ZRTcPkNl7v
[ISOLA24] https://link.springer.com/chapter/10.1007/978-3-031-75387-9_15
[GPCE24] https://dl.acm.org/doi/10.1145/3689484.3690738

**Theoretical Claims:**

N/A

---

> ### Author Rebuttal · Authors · 2025-04-01
>
> Thank you for your very thoughtful review and your positive feedback on our work. We appreciate your recognition of AlphaVerus's potential and thorough understanding of our contributions. We address your points below:
>
> ---
>
> > “However, I have to note that I am not convinced that the rule-based approach can filter out all cases where a specification becomes trivially verifiable. For example assume(false) could be rephrased into something like assume(0!=1)”
>
> The critique modules components work in tandem to collectively filter out bad specifications, so we wouldn’t expect the rule-based filter to filter out all of them. Regarding the case you mentioned (assume(false)), we should clarify that we exclude all assume statements from final proofs, as they are only used for debugging. Therefore, variants like `assume(0 != 1)` will be filtered out. Further, our manual analysis (see response to Reviewer 1) found only 5% critique failiures. That said, we intentionally kept our rules here fairly simple, so expanding this part of the pipeline is interesting future work.
>
> ---
>
> > “Other Comments Or Suggestions”
>
> We thank the reviewer for the suggestions. We will update the algorithm accordingly. Further, to clarify, S from step 1 is only used if no correct solution is found. We will update the algorithm to make it clear.
>
> ---
>
> We also thank the reviewer for sharing additional references that demonstrate the widespread issue of data scarcity in verified code generation. We strongly agree with your perspective and greatly appreciate your acknowledgment that AlphaVerus addresses an important gap. We will appropriately incorporate the valuable citations you provided in our revised manuscript to further strengthen our motivation for AlphaVerus.
>
> ---
>
> We appreciate your valuable feedback and suggestions. We look forward to addressing any additional questions or points you may have.

---

### Official Review · Reviewer_3hti · 2025-03-17

**Overall Recommendation:** 3

**Summary:**

The paper introduces AlphaVerus, a framework for generating formally verified code using LLMs, with focus on the challenges of programming languages with limited training data. AlphaVerus works by translating verified code from programming languages with lots of examples, into the target programming language. First it generates candidate translations, then refines them using a tree search algorithm with code-verifier feedback, and finally filters misaligned specifications and programs. AlphaVerus can generate formally verified solutions for HumanEval-Verified and MBPP-verified.

**Claims And Evidence:**

Claims supported by evidence:

* The paper provides evidence that AlphaVerus improves the translation of programs from Dafny to Verus through more iterations. With experiments, it shows a steady increase in translation success rate.

* The authors show that Treefinement (their refinement approach) increases performance better than additional parallel sampling.

* The critique phase is used to prevent reward hacking.

One question I have is whether the comparison with SAFE and AutoVerus are valid, given they both differ in size and characteristics of datasets and the paper does not try SAFE or AutoVerus on the same datasets as AlphaVerus.

**Essential References Not Discussed:**

-

**Experimental Designs Or Analyses:**

Appendix D states that "critique [...] may not work in all cases, especially for more complex problems". It would be beneficial to expand on this point, if only to provide further characterization of the kinds of complex problems that lead to critique failure.

**Methods And Evaluation Criteria:**

Overall the approach seems reasonable. I wonder whether using Dafny as the source domain limits the applicability, since Dafny is not a mainstream language.

**Other Comments Or Suggestions:**

-

**Other Strengths And Weaknesses:**

-

**Questions For Authors:**

-

**Relation To Broader Scientific Literature:**

-

**Theoretical Claims:**

-

---

> ### Author Rebuttal · Authors · 2025-04-01
>
> Thank you for your thoughtful review of our paper. We're glad you found our approach reasonable and appreciated the evidence supporting AlphaVerus’s effectiveness including treefinement and critique phase to prevent reward hacking. We address your questions and suggestions below:
>
> ---
>
> > One question I have is whether the comparison with SAFE and AutoVerus are valid, given they both differ in size and characteristics of datasets and the paper does not try SAFE or AutoVerus on the same datasets as AlphaVerus.
>
> We do think the comparisons are valid, so let us make a few clarifications. First, the comparison in Table 4 uses the exact same benchmark dataset: MBPP-verified. SAFE++ and AutoVerus were specifically designed for the proof annotation task, so we used AlphaVerus on this task to give the baselines the strongest possible chance. Regarding SAFE++, the authors have not released their code, so we were limited to reporting their published numbers.
>
> For the verified code generation tasks, we attempted to adapt the publicly available AutoVerus framework, but despite significant effort, could not achieve non-trivial performance. As we note in the paper, this is almost certainly due to their task-specific, hand-engineered prompts.
>
> Finally, we note that this research area is still emerging, and even SAFE++ and AutoVerus are fairly concurrent works, so we made our best effort to construct a fair comparison given their results and models. We would be happy to further clarify any other aspects of this evaluation.
>
> ---
>
> > I wonder whether using Dafny as the source domain limits the applicability, since Dafny is not a mainstream language.
>
> Our goal in AlphaVerus is to perform formally verified code generation. Among automated program verification languages, Dafny is one of the most prominent in terms of code content and industrial use, including large-scale deployments [1].
>
> Further, our primary goal was demonstrating the *feasibility of the bootstrapping pipeline* (translation -> refinement -> critique -> self-improvement) in a realistic data-scarce setting for verified code. Dafny was chosen strategically because it *has* accumulated sufficient verified examples over time to serve as a viable source, unlike many target verification languages (like Verus). Crucially, our pipeline makes *minimal assumptions about the source language* (e.g., no source verifier needed). The core methodology – leveraging a higher-resource language, iterative refinement with verifier feedback, and critique – is applicable to other language pairs where a similar resource disparity exists.
>
> ---
>
> > Appendix D states that "critique [...] may not work in all cases, especially for more complex problems". It would be beneficial to expand on this point...
>
> This statement in the limitations (Appendix D) was included in the spirit of acknowledging *potential future challenges* as problem complexity scales significantly beyond current benchmarks. However, within our experiments on DafnyBench translation and HumanEval/MBPP-verified generation, we did not observe failure patterns where the critique allowed misaligned or trivially verified programs through. That said, identifying specific limitations of the critique component and improving it further would be an interesting direction for future work to explore.
>
> ---
>
> We hope these clarifications address your concerns. We value your feedback and believe these points strengthen our contribution. We look forward to addressing any additional questions or points you may have.
>
> ## References:
> [1] Chakarov, Aleks, et al. Formally Verified Cloud-Scale Authorization. 2025, https://www.amazon.science/publications/formally-verified-cloud-scale-authorization.

---

### Official Review · Reviewer_5hh1 · 2025-03-17

**Overall Recommendation:** 4

**Summary:**

The paper introduces a novel (ensemble of) technique(s) for the translation and formal verification of programs using Verus, a library based on the Rust language.
The authors implement a 3 step process that utilizes an LLM to generate samples of a program translation and proof of formal verification, a tree based search to refine candidates that presents errors in the translation, and a final LLM critique step to test whether the specifications of the original program have been respected in the translation process and the formal proof has been conducted correctly. By iteratively applying this process, the successful translations are collected in a dataset to be used as in context examples for future iterations, leading to an incremental improvement in translation performance.

The novel contributions are in the combined usage of the tree based search (Treefinement) and the critique actor; the latter include very language specific specifications (string matching) but also more generalizable methods and looks promising to be used in other context as well.

## update after rebuttal
Nothing really changed with the rebuttal. I believe the paper should be accepted.

**Claims And Evidence:**

The claims seems to be well supported by the ablation studies showcased in the result and appendix sections. Treefinement is a win as it allows (in combination of exploration) to improve the results to best in class. The critique actor is also needed as showcased in figure 7. The idea is that without it the translations tend to degenerate in hacked solution rather than valid ones.

**Essential References Not Discussed:**

N/A

**Experimental Designs Or Analyses:**

Experimental designs are sound and present a fair comparison of this methodology with baselines. As this is the first paper addressing translation and formal verification in Verus, benchmarks rely on the few-shots techniques using accessible and well performing LLMs, but not on other specialized prompting and verification strategies.

It seems we just trust the critique in the final evaluation of translation.  While the reward hacking analysis is nice, it's not clear yet whether we can fully trust the generated specs.

**Methods And Evaluation Criteria:**

Yes, the evaluation criteria of a successful translation and verification is sensible. The scope is limited to datasets with human labeled and verified translations to Verus, which are limited in size.

**Other Comments Or Suggestions:**

I think the section on the critique could use a bit more exposition in the main paper, though the detail of the appendix is appreciated.

**Other Strengths And Weaknesses:**

The paper presents an interesting methodology to kick-start and augment datasets for translation and formal verification of programs. The feedback from the compiler during Treefinement looks particularly useful to generate sound proofs that pass the compiler test and thus are free of potential hallucinations, increasing the probability of the creation of valid of the code, especially in the early phases when an LLM could have trouble to generate valid proofs in the absence of data. The critique is also particularly useful to avoid the obvious reward hacking. However, it is unclear how scalable this last step is to other languages: it will require different human coded rules, and the peculiarity of other languages might also make reward hacking harder to identify.

The self-improvement mechanism using in-context examples is interesting and is a good way of getting self improvement without being able to fine tune a model.  That being said, its not clear if this approach is limited over methods that also improve the model weights as well.  It was interesting seeing the exemplars translate to other models.

**Questions For Authors:**

How can you evaluate the critique model in a general and scalable way.  It seems like the reward hacking analysis focuses on one particular hack, but I imagine there is no shortage of other hacks.

Can you subsample proofs and manually verify to get numbers that can be trusted.

**Relation To Broader Scientific Literature:**

This paper is relevant to work at the intersection of formal verification and LLMs.  Bootstrapping off one formal language to another is a common difficulty with data scarcity in the space. Beyond rust, this provides a promising method for using a lot of code.  I'd be curious if this works in math with isabelle proofs -> lean proofs or similar.

**Theoretical Claims:**

N/A

---

> ### Author Rebuttal · Authors · 2025-04-01
>
> Thank you for your thoughtful review and positive assessment of our work. We appreciate your recognition of AlphaVerus’s novelty, useful and interesting methodology, particularly the Treefinement and Critique components, while finding it a promising method for other contexts. We are happy to address your questions and suggestions below.
>
> ---
>
> > Yes, the evaluation criteria of a successful translation and verification is sensible. The scope is limited to datasets with human labeled and verified translations to Verus, which are limited in size.
>
> Thank you for the comments about sensible evaluation criteria. Regarding scope, we agree that current evaluation datasets (HumanEval-Verus, MBPP-verified, Sec 4) are limited. This is primarily because the research direction of formally verified code generation is still emerging, and verifying code is extremely difficult even for human experts. We would be excited if future work expands the scope of benchmarks. Aside from evaluation, we note that our pipeline makes limited assumptions about the underlying data, so we also see adapting AlphaVerus to different source domains as an exciting future direction.
>
> > It seems we just trust the critique in the final evaluation of translation. While the reward hacking analysis is nice, it's not clear yet whether we can fully trust the generated specs.
>
> Thank you for raising this interesting point about the generated specifications. Indeed, ensuring that the generated specification aligns with the source intent is challenging, as specs lack formal guarantees relative to intent. In our case, we evaluated AlphaVerus on downstream tasks in which the specifications are human-written. This means that the generated specifications from the AlphaVerus pipeline were at least useful for improving verified code generation performance.
>
> In addition, motivated by your feedback, we conducted a manual review on a random subset of 20 examples and found 95% to be correct and complete translations, suggesting that the AlphaVerus pipeline frequently produces reasonable specifications. However, fully trusting AI-generated specs is an open and challenging area for future work to explore.
>
> > The critique is also particularly useful to avoid the obvious reward hacking. However, it is unclear how scalable this last step is to other languages: it will require different human coded rules, and the peculiarity of other languages might also make reward hacking harder to identify.
>
> Although rules can vary by language, our critique stage includes components (comparison and exploit models) that make minimal assumptions about the target language. The rule-based component is also intentionally simple, involving basic pattern matching (e.g., checking for assume statements, trivial preconditions, and verifier bypass annotations). Most of these would carry over to other verification-oriented languages. Nonetheless, fully adapting AlphaVerus to other languages remains an interesting future direction.
>
> > I think the section on the critique could use a bit more exposition in the main paper, though the detail of the appendix is appreciated.
>
> Thank you for highlighting this. We will incorporate a clearer and more detailed summary of the critique step directly into the main paper.
>
> ---
>
> We hope these responses address your points. We appreciate the encouraging feedback and believe these clarifications will strengthen the final paper. We are happy to discuss any further concerns.

---

### Decision · Program_Chairs · 2025-05-01

**Decision:**

Accept (poster)

**Comment:**

This paper proposes AlphaVerus, a method to augment the data for verified code generation by iteratively translating programs from high-resource to low-resource programming languages. After the exploration of translation, it subsequently does treefinement to search for program refinement using verifier feedback and filtering out misaligned programs to prevent reward hacking. Although the individual steps proposed in this work might not be novel (e.g., hi-res -> low-res translation, tree search, etc), the combination of use of language models and formal verification for generating verified code is interesting and novel. And more broadly, efforts in connecting the fields of programming language and machine learning research is always appreciated. The experiments are also well-constructed, as three LLMs and 2 programming datasets are used to show the effectiveness of the proposed method. And the specific evaluation used in this work (i.e., model must generate both the code and the proof annotations) is refreshing. Overall, I recommend an acceptance to the conference for this paper.